# Multi-Mode Channel Position Attention Fusion Side-Scan Sonar Transfer Recognition

Jian Wang [1,2,3], Haisen Li [1,2,3,*], Guanying Huo [4], Chao Li [1,2,3] and Yuhang Wei [1,2,3]

1. Acoustic Science and Technology Laboratory, Harbin Engineering University, Harbin 150001, China
2. College of Underwater Acoustic Engineering, Harbin Engineering University, Harbin 150001, China
3. Key Laboratory of Marine Information Acquisition and Security, Harbin Engineering University, Ministry of Industry and Information Technology, Harbin 150001, China
4. College of Internet of Things Engineering, Hohai University, Changzhou 213022, China
* Correspondence: hsenli@126.com

**Abstract:** Side-scan sonar (SSS) target recognition is an important part of building an underwater detection system and ensuring a high-precision perception of underwater information. In this paper, a novel multi-channel multi-location attention mechanism is proposed for a multi-modal phased transfer side-scan sonar target recognition model. Optical images from the ImageNet database, synthetic aperture radar (SAR) images and SSS images are used as the training datasets. The backbone network for feature extraction is transferred and learned by a staged transfer learning method. The head network used to predict the type of target extracts the attention features of SSS through a multi-channel and multi-position attention mechanism, and subsequently performs target recognition. The proposed model is tested on the SSS test dataset and evaluated using several metrics, and compared with different recognition algorithms as well. The results show that the model has better recognition accuracy and robustness for SSS targets.

**Keywords:** side-scan sonar image classification; attention mechanism; multi-modal transfer learning; multi-channel; synthetic aperture radar

## 1. Introduction

Side-scan sonar (SSS) generates pseudo-color images of varying intensities by recording the intensity of backscattered sound waves from the ocean floor. It has the characteristics of wide coverage and high resolution and can not only map seabed topography but also image underwater targets (wrecks, aircraft). It is the main sensor used in autonomous underwater vehicles (AUV). In recent years, automatic target recognition (ATR) based on SSS data is used to quickly and automatically classify underwater targets [1]. Real-time detection and identification of submarine objects using low-cost and low-risk AUVs is an issue of great interest to Navies around the world [2,3]. However, the signal-to-noise ratio of sonar images is extremely low, and the target information is seriously polluted due to the existence of ocean noise, reverberation noise and speckle noise [4–7]. In addition, with the increase of the operating frequency of SSS, the absorption of sound energy by seawater increases. Consequently, the sound absorption loss of sound waves in seawater is large and the sonar images exhibit the low target contrast phenomenon [8,9], which directly affects the target recognition performance.

The deep learning (DL) technology can convolve the input image through multiple convolution kernels of different scales, filtering the image and realizing the feature extraction of the input image. The focus of a deep learning object recognition system is to design the structure of the convolutional neural network. The input image is extracted from shallow to deep through the designed network structure, and the target is identified by the classifier according to the features extracted by the final layer. It can be observed that for the target recognition method based on the convolutional neural network (CNN) [10–13], the

most important factor is to extract stable features. The instability of features can be reflected in the problems of insufficient data samples and features susceptible to noise interference. If the designed network is huge and the network parameters have high magnitudes, the recognition network will have an overfitting problem [14]. This means that the recognition performance of the network on the training and test sets will be very different, and the features extracted under noise interference will have noisy features. Such features have a negative impact on final recognition. Therefore, in order to obtain a good SSS target recognition performance, it is necessary to mitigate the problem associated with insufficient training samples and study the generation of stable features in the network design process.

### 1.1. Literature Review

Recent research on SSS target recognition has shown that the target recognition method based on CNNs has outperformed traditional machine learning methods, including the fuzzy logic method, K nearest neighbor, support vector machines, etc., [15–17]. With an increasing scale, the recognition network can extract deeper features from images to obtain richer feature information. Therefore, a research direction for target recognition is to design a deeper recognition network model. Simonyan et al., Alex et al., He et al. and Xu et al. proposed VGG [18], AlexNet [19], ResNet [20], and DesNet [21] methods, respectively. On the basis, the ResNet152 and Desnet161 methods were proposed to increase the number of network layers and improve the ability of feature extraction [22,23]. However, the aforementioned methods require a large number of data samples in the process of parameter learning. As it is difficult to obtain sample data for SSS data, the lack of training samples limits the application of the methods proposed in [20–23] for SSS target recognition. In contrast, sample transfer learning (STL) adopts the method of source data set to train the main network body of the model, and target data set to fine-tune the head network, which can effectively solve the problem of insufficient training samples. This research direction has received a significant amount of attention recently. For example, Ye et al. applied the VGG-11 and ResNet-18 models to the underwater target identification of SSS images and adopted optical data pre-training and SSS data fine-tuning to improve the low recognition rate caused by insufficient data samples [24]. Huo et al. used semi-synthetic and SSS data to fine-tune the VGG parameters in order to further improve the sample identification accuracy [25]. Tang et al. detected shipwreck targets using SSS based on the trained DarkNet-53 network [26]. Fuchs et al. applied the transfer recognition method to the target recognition of forward-looking sonar (FLS) [27]. Zhang et al. adopted the YOLOv5 model to detect FLS targets [28].

All the above studies have proved the effectiveness of STL in overcoming the problem of insufficient samples. These studies use different data sets and target data sets to pre-train different backbone networks and fine-tune the head network according to different task requirements, respectively, and finally achieve detection and recognition. However, the SSS images have strong noise interference that is considerably different from the optical images. This difference in distribution causes the features learned by the pre-trained network to significantly limit the recognition accuracy of the head network. In order to overcome this shortcoming, we should study the use of different modal data sets to train the network in different stages, so as to alleviate the problem of insufficient recognition accuracy caused by data differences.

Zhen et al. used synthetic aperture radar (SAR) data to train the middle layer of the network. At the same time, the attention mechanism was utilized to make the features extracted by the network more representative, which improved the anti-noise performance and network recognition accuracy [29]. Although this method was effective, the attention weights of different positions of the same channel were the same in the attention network design. Furthermore, the attention weights of different channels of the same position were also the same. This network design reduced the attention diversity in the network. To address the aforementioned limitations, it is necessary to carry out research on multi-channel multi-location attention networks.

*1.2. Novel Contributions*

Aiming at the above research gaps, a new multi-channel multi-location attention model (MMA) is proposed for SSS target recognition. In this model, the ResNet network is used as the backbone network, and the ImageNet dataset and the SAR dataset are used to train different stages of the backbone network to improve its data adaptability to SSS. Finally, the MMA model and head recognition network are trained using the SSS dataset to produce accurate and robust recognition results. The novel scientific contributions of this study are as follows:

1. A deep learning model for SSS object recognition is proposed. The model structure can learn the backbone network parameters in stages through a variety of data sets, reduce the distribution difference between the ImageNet data and SSS data, and improve the recognition accuracy for SSS targets. The effectiveness of the proposed recognition model is verified on a measured SSS target dataset (https://toscode.gitee.com/wangjian1987011 8/ssd-dataset.git, accessed on 27 January 2023).

2. In the previous research on transfer target recognition, only the fully connected layer of the head network was transferred and adjusted, which limited the learning scope. There was no key feature extraction of the key positions and key channels of the SSS target. Unlike previous research, a multi-channel and multi-position attention mechanism is proposed. During the process of extracting SSS target features, different channel attention factors can be set for channels at different positions, and positions can be set at different channel attention factors. These diverse attention factors can achieve all-round acquisition of SSS target features.

3. An integration strategy based on multimodal staged transfer is proposed to enhance the generalization performance of the model. Unlike the existing research, this strategy can enable the backbone network to simultaneously have better classification accuracy for multiple different modal data, so that the SSS target recognition has better recognition performance.

## 2. Methods

The multi-modal multi-channel multi-position attention transfer recognition model (3MATR) proposed in this paper consists of two stages: (1) multi-modal staged transfer stage and, (2) SSS image attention recognition stage. Figure 1 shows the specific framework of the proposed model.

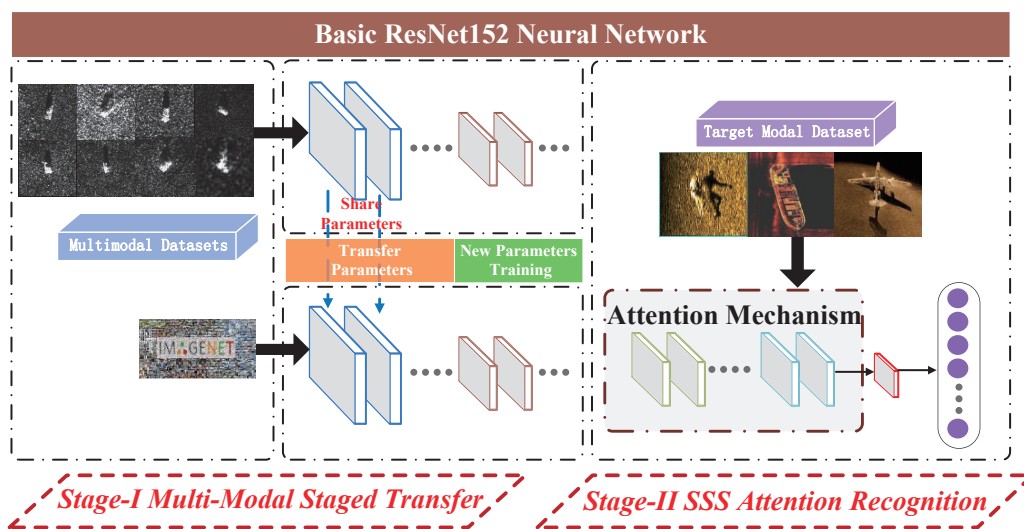

**Figure 1.** Framework of the proposed 3MATR model.

1. Stage-I: multi-modal staged transfer.

The training data set is divided into an optical ImageNet data set and the SAR data set. The latter is used to train the front part of the backbone network (ResNet152), and the former is used to train the middle part of the network. This stage enables the backbone network to accurately identify noisy datasets.

2. Stage-II: multi-channel multi-location attention model target recognition.

The backbone network learned from multiple datasets is combined with the attention mechanism network (AMN) and the recognition network to build an integrated learning model. The SSS data are used to learn the parameters of the network at this stage. During the parameter learning process, attention is allocated to different positions of different channels to obtain the important features of the target. Finally, the SSS target is accurately identified.

### 2.1. Multi-Modal Staged Transfer

The image size of the training data set is 224 × 224, and the backbone network adopts the ResNet series network structure. Figure 2 shows the local structure. The constant mapping in the ResNet structure can solve the vanishing gradient problem and model degradation during the parameter learning process. In this study, the SAR and ImageNet datasets are used to train the front and middle sections of ResNet152, respectively. The ResNet152 network consists of multiple 3 × 3 and 1 × 1 convolution kernels. The former kernel unit extracts the features of the image information, and the latter kernel can upgrade and reduce the dimensionality of the feature channel through the integration of cross-channel features. Figure 3 shows the ResNet152 network structure diagram.

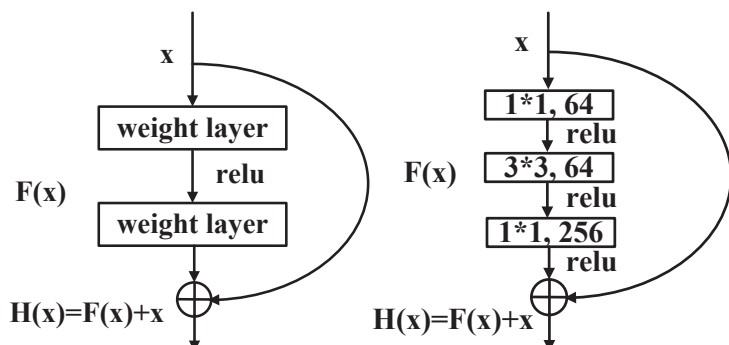

**Figure 2.** Resnet local operation structure diagram.

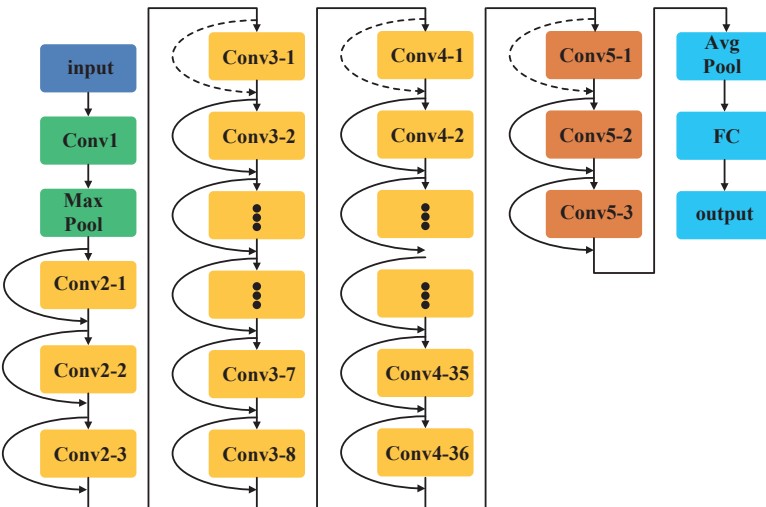

**Figure 3.** Detailed architecture diagram of the ResNet152 network.

The ResNet152 model is widely used in target recognition problems. The backbone network is composed of five parts: $Conv1$, $Conv2\_x(x = 1\ldots3)$, $Conv3\_x(x = 1\ldots8)$, $Conv4\_x(x = 1\ldots36)$, and $Conv5\_x(x = 1\ldots3)$, where $Conv1$ and $Conv2\_x(x = 1\ldots3)$ form the front segment of the network using SAR data for parameter learning, and $Conv3\_x$, and $Conv4\_x$ form the middle segment of the network using ImageNet data for parameter learning. $Conv5\_x(x = 1\ldots3)$ and the remaining parts of the network use SSS data for transfer parameter learning. In this way, the purpose of phased transfer learning is achieved. Table 1 shows the details of the convolution kernel settings in ResNet152.

**Table 1.** Resnet152 network parameter settings.

| Layer Name | Output Size | Resnet152 |
|:---:|:---:|:---:|
| $conv1$ | $112 \times 112$ | $7 \times 7.64, s = 2$ |
| $conv2\_x(x = 1\ldots3)$ | $56 \times 56$ | $3 \times 3$ max pool, $s = 2$<br>$1 \times 1.64$<br>$3 \times 3.64$<br>$1 \times 1.256$ |
| $conv3\_x(x = 1\ldots8)$ | $28 \times 28$ | $1 \times 1.128$<br>$3 \times 3.128$<br>$1 \times 1.512$ |
| $conv4\_x(x = 1\ldots36)$ | $14 \times 14$ | $1 \times 1.256$<br>$3 \times 3.256$<br>$1 \times 1.1024$ |
| $conv5\_x(x = 1\ldots3)$ | $7 \times 7$ | $1 \times 1.512$<br>$3 \times 3.512$<br>$1 \times 1.2048$ |
| Output | $1 \times 1$<br>1000-d | average pool<br>fc, softmax |

### 2.2. Multi-Channel Multi-Location Attention Model

The convolution operation in the network uses convolution kernels of different sizes to extract features of local areas, therefore, it cannot perceive the entire position space. A larger position space can be perceived by using a deeper network structure. Although this method perceives the entire location space, it splits the spatial information and can easily lead to a loss of feature information. In this paper, we propose a method that can collect global features and preserve positional relationships. This method can help subsequent convolutional layers perceive the entire space and capture complex feature relationships. At the same time, we propose a multi-channel and multi-location attention mechanism, which can fully consider the channel attention factors of different positions and the position attention factors of different channels. In this way, compact key features can be obtained from complex global images, and the importance and robustness of the proposed features can be enhanced. Next, we describe the architecture and theory of the proposed method in detail.

#### 2.2.1. Channel Attention Mechanism Considering Location (CAMCL)

The design of the CAMCL attention model proposed in this paper is based on the QKV model [30–33] in the self-attention mechanism. Let the input image be $X \in \mathbb{R}^{c \times h \times w}$, denoting the input tensor of a spatiotemporal (2D) convolutional layer. The number of channels is represented by $c$, and $h$ and $w$ represent the height and width of the spatial

scale, respectively. The feature map of $X$ through the convolution kernel is denoted by $A, B, V$, where $A, B, V$ are three different layers represented as follows:

$$
\begin{aligned}
A &= \phi(X; W_\phi) \\
B &= \theta(X; W_\theta) \\
V &= \varphi(X; W_\varphi)
\end{aligned}
\tag{1}
$$

where $W_\phi$, $W_\theta$ and $W_\varphi$ are the corresponding network parameters. The layers $A, B$ and $V$ are expanded in the spatial dimension, where $B^{c \times hw} = [b_1, b_2, \ldots b_{hw}]$ and $V^{c \times hw} = [v_1, v_2, \ldots v_{hw}]$. In order to calculate the channel attention factor, this paper uses the matrix product $AB^T$ to calculate the channel correlation matrix $G^{c \times c}$, which is described as

$$
G^{c \times c} = [g_1, g_2, \ldots g_c] = AB^T = [a_1, a_2, \ldots a_{hw}][b_1, b_2, \ldots b_{hw}]^T = [a_1, a_2, \ldots a_{hw}]
\begin{bmatrix}
b_1{}^T \\
b_2{}^T \\
\vdots \\
b_{hw}{}^T
\end{bmatrix}
$$

$$
=
\begin{bmatrix}
\overline{a_1} \\
\overline{a_2} \\
\vdots \\
\overline{a_c}
\end{bmatrix}
[\overline{b_1}, \overline{b_2}, \ldots \overline{b_c}]
=
\begin{bmatrix}
\overline{a_1}\overline{b_1} & \overline{a_1}\overline{b_2} & \cdots & \overline{a_1}\overline{b_c} \\
\overline{a_2}\overline{b_1} & \overline{a_2}\overline{b_2} & \cdots & \overline{a_2}\overline{b_c} \\
\vdots & \vdots & \cdots & \vdots \\
\overline{a_c}\overline{b_1} & \overline{a_c}\overline{b_2} & \cdots & \overline{a_c}\overline{b_c}
\end{bmatrix}
\begin{array}{l}
\overline{a_i} \in {}^{1 \times hw} \overline{b_j} \in {}^{hw \times 1} \\
i, j = 1, 2, \cdots c
\end{array}
\tag{2}
$$

$$
\overline{a_i}\overline{b_j}
$$

where $\overline{a_i}\overline{b_j}$ indicates the attention factor between channels $i$ and $j$. The image is processed by channel attention, which can be expressed as $Z^{c \times hw} = G^{c \times c} V^{c \times hw}$. The image after channel attention processing is obtained by reshaping $Z$ in order to take into account the position factor while performing channel attention. In Equation (2), $\overline{b_j}(j = 1, \ldots, hw)$ is transformed into an effective position attention weight vector using $softmax()$. At the same time, $V^{c \times hw} = [\overline{v_1}; \overline{v_2}, \ldots, \overline{v_c}]$ also performs the transformation identical to $A = [\overline{a_1}; \overline{a_2}, \ldots \overline{a_c}]$. Therefore, using Equation (2), the channel attention mechanism considering the location factor can be expressed as follows:

$$
\widetilde{Z}^{c \times hw} = \widetilde{G}^{c \times c} \widetilde{V}^{c \times hw}
$$

$$
=
\begin{bmatrix}
\overline{a_1}soft\max(\overline{b_1}) & \overline{a_1}soft\max(\overline{b_2}) & \cdots & \overline{a_1}soft\max(\overline{b_c}) \\
\overline{a_2}soft\max(\overline{b_1}) & \overline{a_2}soft\max(\overline{b_2}) & \cdots & \overline{a_2}soft\max(\overline{b_c}) \\
\vdots & \vdots & \cdots & \vdots \\
\overline{a_c}soft\max(\overline{b_1}) & \overline{a_c}soft\max(\overline{b_2}) & \cdots & \overline{a_c}soft\max(\overline{b_c})
\end{bmatrix}
\begin{bmatrix}
soft\max(\overline{v_1}) \\
soft\max(\overline{v_2}) \\
\vdots \\
soft\max(\overline{v_c})
\end{bmatrix}
$$

$$
\begin{array}{l}
\overline{a_i} \in {}^{1 \times hw} \overline{b_j} \in {}^{hw \times 1} \overline{v_k} \in {}^{1 \times hw} \\
i, j, k = 1, 2, \cdots c
\end{array}
\tag{3}
$$

Figure 4 shows the calculation block diagram of the channel attention mechanism considering the positions.

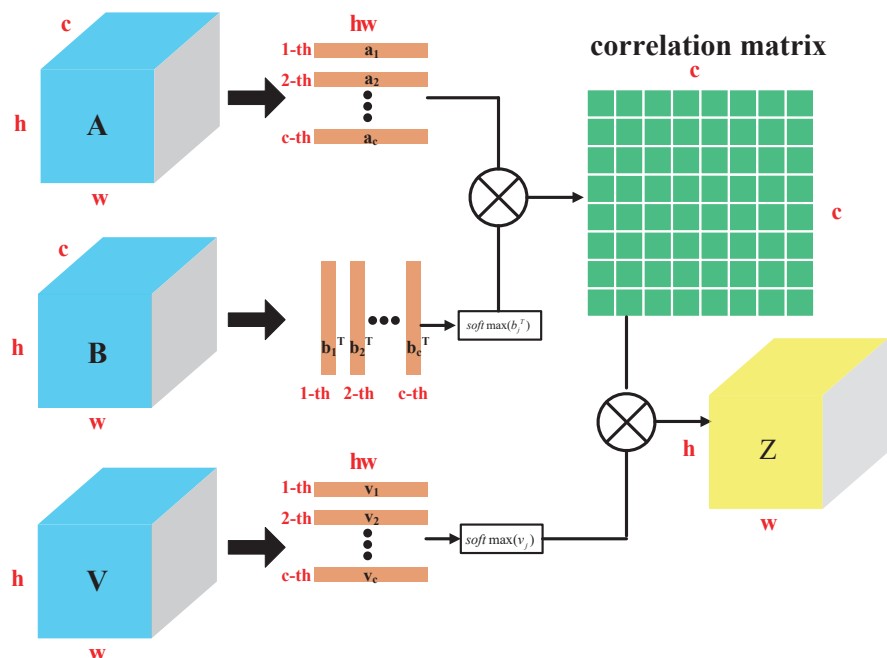

**Figure 4.** CAMCL calculation flow chart.

2.2.2. Location Attention Mechanism Considering Channel (LAMCC)

In order to calculate the location attention factor, this paper uses the matrix multiplication $A^T B$ to calculate the location correlation matrix $F^{hw \times hw} = [f_1, f_2, \ldots, f_{hw}]$ as follows:

$$
\begin{aligned}
F^{hw \times hw} &= [f_1, f_2, \ldots, f_{hw}] = A^T B = [a_1, a_2, \ldots, a_{hw}]^T [b_1, b_2, \ldots, b_{hw}] \\
&= \begin{bmatrix} a_1{}^T \\ a_2{}^T \\ \vdots \\ a_{hw}{}^T \end{bmatrix} [b_1, b_2, \ldots, b_{hw}] = \begin{bmatrix} a_1{}^T b_1 & a_1{}^T b_{hw} & \cdots & a_1{}^T b_{hw} \\ a_2{}^T b_1 & a_2{}^T b_{hw} & \cdots & a_2{}^T b_{hw} \\ \vdots & \vdots & \cdots & \vdots \\ a_{hw} b_1 & a_{hw} b_{hw} & \cdots & a_{hw} b_{hw} \end{bmatrix} \\
& a_i \in \mathbb{R}^{c \times 1} b_j \in \mathbb{R}^{c \times 1} \\
& i, j = 1, 2, \cdots hw
\end{aligned}
\tag{4}
$$

where $a_i^T b_j$ indicates the attention factor between locations $i$ and $j$. The location attention operation of an image can be expressed as $K^{hw \times c} = F^{hw \times hw}(V^{c \times hw})^T$, which can be reshaped to obtain the image after location attention processing. In order to consider the channel attention factor while performing position attention, we adopt an idea similar to CAMCL, and express Equation (4) as follows using the $softmax()$ function:

$$
\begin{aligned}
\widetilde{K}^{hw \times c} &= \tilde{F}^{hw \times hw} \left( \tilde{V}^{c \times hw} \right)^T \\
&= \begin{bmatrix} a_1{}^T softmax(b_1) & a_1{}^T softmax(b_2) & \cdots & a_1{}^T softmax(b_{hw}) \\ a_2{}^T softmax(b_1) & a_2{}^T softmax(b_2) & \cdots & a_2{}^T softmax(b_{hw}) \\ \vdots & \vdots & \cdots & \vdots \\ a_{hw}{}^T softmax(b_1) & a_{hw}{}^T softmax(b_2) & \cdots & a_{hw}{}^T softmax(b_{hw}) \end{bmatrix} \begin{bmatrix} softmax(v_1)^T \\ softmax(v_2)^T \\ \vdots \\ softmax(v_{hw})^T \end{bmatrix} \\
& a_i \in \mathbb{R}^{c \times 1} b_j \in \mathbb{R}^{c \times 1} v_k \in \mathbb{R}^{c \times 1} \\
& i, j, k = 1, 2, \cdots hw
\end{aligned}
\tag{5}
$$

Figure 5 shows the calculation block diagram of the location attention mechanism considering the channel.

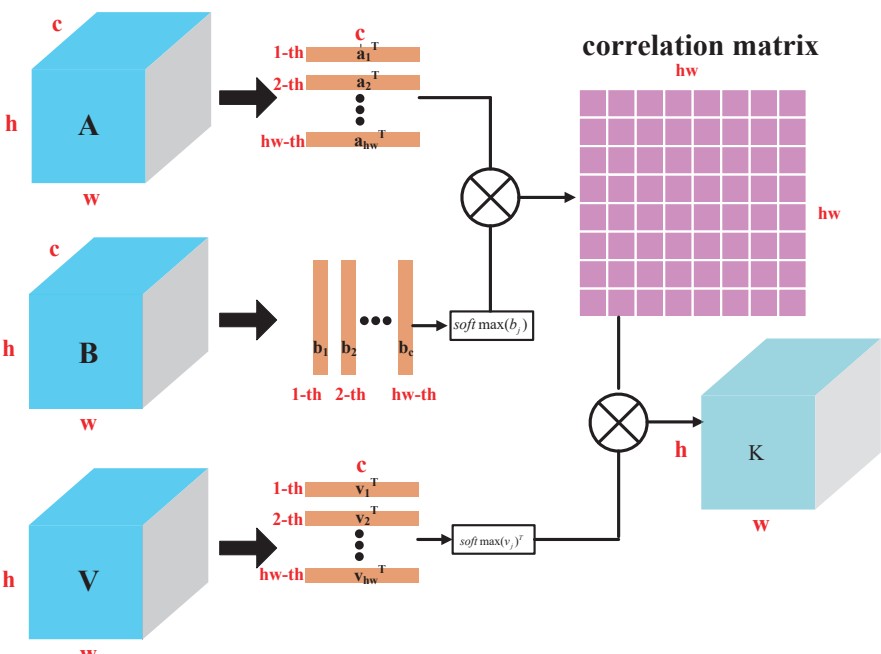

**Figure 5.** LAMCC calculation flow chart.

### 2.3. Data Description

#### 2.3.1. SSS Dataset

A target image dataset collected via imaging sonar is used to conduct the experiments. The dataset used in the experiment includes three types of image targets as follows: 66 planes, 484 ships, and 578 other pictures. Figure 6 shows some example dataset images. It can be observed that the image has strong noise interference, which causes considerable difficulty in accurate target identification.

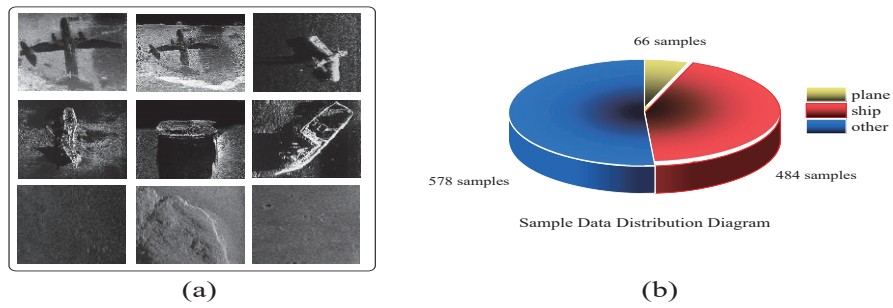

(a)                                          (b)

**Figure 6.** Side-scan sonar dataset samples. (**a**) Three classes of side-scan image targets; (**b**) Sample distribution diagram.

#### 2.3.2. Experimental Dataset Preprocessing

The data are randomly extracted for creating training and test datasets, where 70% of the data are used as training data and the remaining 30% are used as test data. Table 2 shows the specific data allocation for each category. In order to reduce the influence of random initialization of parameters on the recognition performance, values obtained over repeated experiments are averaged and used for the final evaluation of the recognition performance. The data types in the dataset are unbalanced, with the largest being the 578 samples of other target types and only 66 plane samples. The unbalanced data will cause the classifier to favor categories with large sample sizes, and the recognition rates of the small sample categories will be poor. At the same time, the total number of samples in the data set is 1128, and the network trained using this dataset will exhibit overfitting.

**Table 2.** Specific data allocation for each category.

| Categories / Numbers | Plane | Ship | Other |
|---|---|---|---|
| Total | 66 | 484 | 578 |
| After dataset division (train set 70%, test set 30%) | | | |
| Train | 46 | 338 | 404 |
| Test | 20 | 146 | 174 |

In this experiment, in order to reduce the impact of unbalanced data and overfitting on the evaluated algorithms, some basic data augmentation methods (flip, rotation, cropping) are also applied to preprocess the data. These include center crop, left bottom crop, left top crop, right bottom crop, right top crop, equal height stretch, equal width stretch, contrast transformation, rotation (45°, 90°, 135°, 180°, 225°, 270°, 315°), left and right flip. Figure 7 shows the image transformation result and Figure 8 shows the image data used in the experiment. It can be observed from the figures that different types of images have different appearances and distributions.

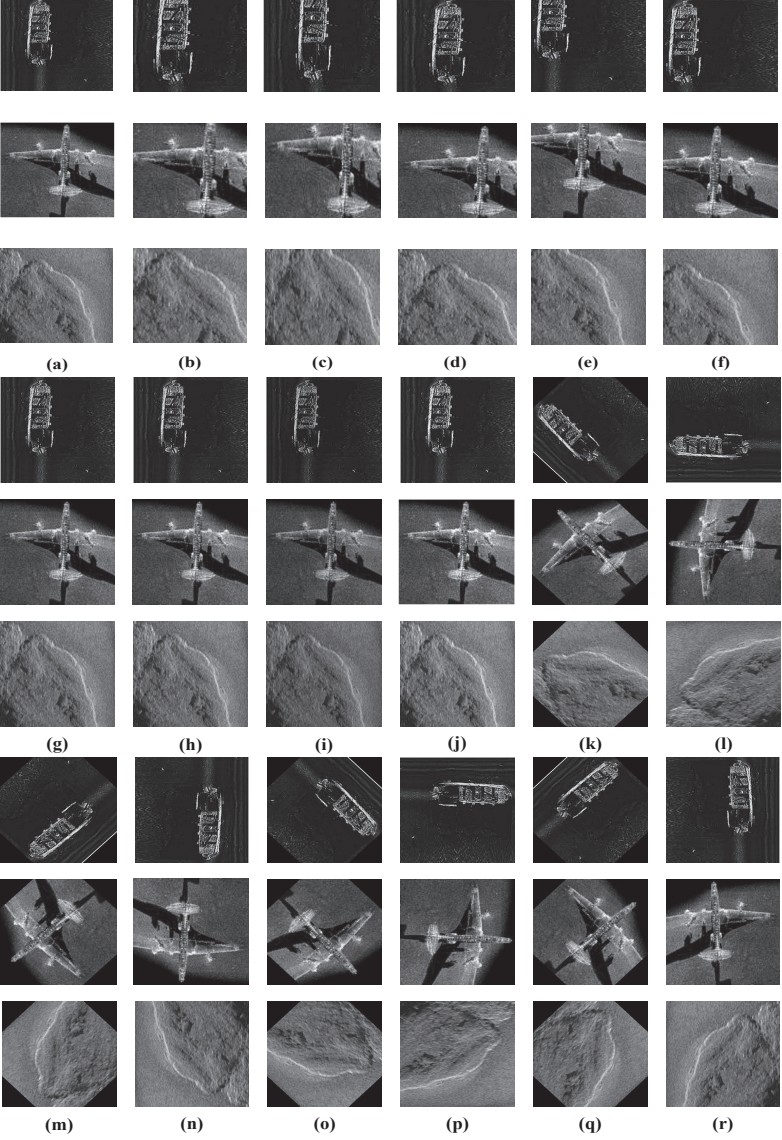

**Figure 7.** Image transformation result graph: (**a**) image sample; (**b**) center crop; (**c**) left bottom crop; (**d**) left top crop; (**e**) right bottom crop; (**f**) right top crop; (**g**) equal height stretch; (**h**) equal width stretch;

(**i**) contrast transformation gamma = 0.87; (**j**) contrast transformation gamma = 1.07; (**k**) rotation = 45°; (**l**) rotation = 90°; (**m**) rotation = 135°; (**n**) rotation = 180°; (**o**) rotation = 225°; (**p**) rotation = 270°; (**q**) rotation = 315°; (**r**) left and right flip.

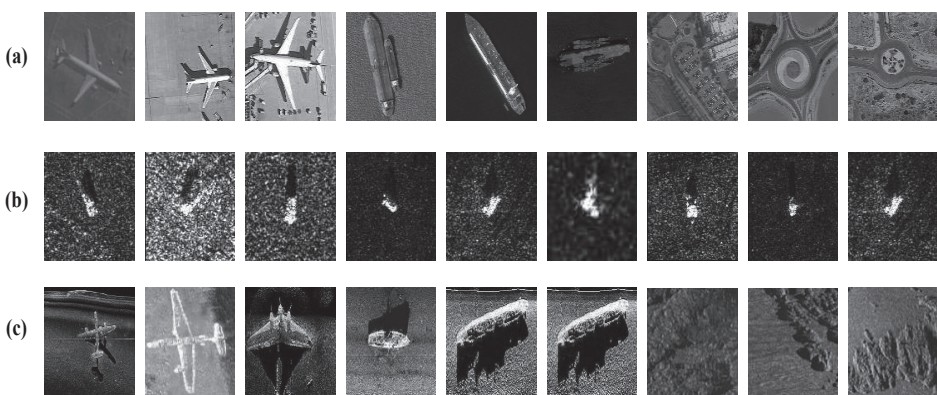

**Figure 8.** Datasets used in the experiments: (**a**) Grayscale optical image samples; (**b**) SAR image samples; (**c**) SSS image samples.

### 2.4. Evaluation Metric

We use accuracy to evaluate the criterion for evaluating the model performance. The training network in this paper uses the ResNet152 network. The network uses SAR images and optical images to train the front section and back end of the backbone network, respectively, and uses SSS images to train the head network. The batch size is 30, the learning rate is 0.001, and model validation is performed 10 times using the cross-validation method, The criterion for assessing model performance is the average overall accuracy (OA), which is the percentage of all correct positive classifications and represents the overall classification performance. The OA is calculated as in Equation (6).

$$\mathrm{OA} = \frac{\sum_i^C N_{ii}}{N} \tag{6}$$

where $N_{ii}$ is the number of test samples that should have been classified as class $i$ and are classified as class $i$ in the actual classification results, $c$ refers to the categories of labels in the test samples, and $N$ is the total number of test samples.

### 2.5. Experimental Platform

Experiments are run on a Microsoft Windows 10 operating system with an NVIDIA GTX TITAN-XP GPU and 64 GB of memory. Python 3.6.8 version is used to design the network structure.

## 3. Results and Discussion

In this part, the robustness and effectiveness of the proposed method are verified through comparative experiments and analysis. The method is compared with traditional DL recognition methods to verify the improvement effect of the algorithm on the recognition rate. In addition, this algorithm is compared with related transfer learning (TL) algorithms.

The OA index of the test data set is used to compare the state-of-the-art (SOTA) methods with the method proposed in this paper and analyze the related performance of the results. At the same time, data ablation experiments, CAMCL and LAMCC ablation experiments are used to analyze the performance of the network structure, and the method proposed in this paper is verified using the forward-looking sonar (FLS) data set for sample diversity.

(1) Comparison of our method with traditional DL models

DenseNet, ResNet, VGGNet and other series methods are the most classical and widely used network structures for target recognition. Therefore, this paper chooses these

methods for comparative analysis and lists their details and performance. Experimental results are shown in Figure 9 and Figure 10.

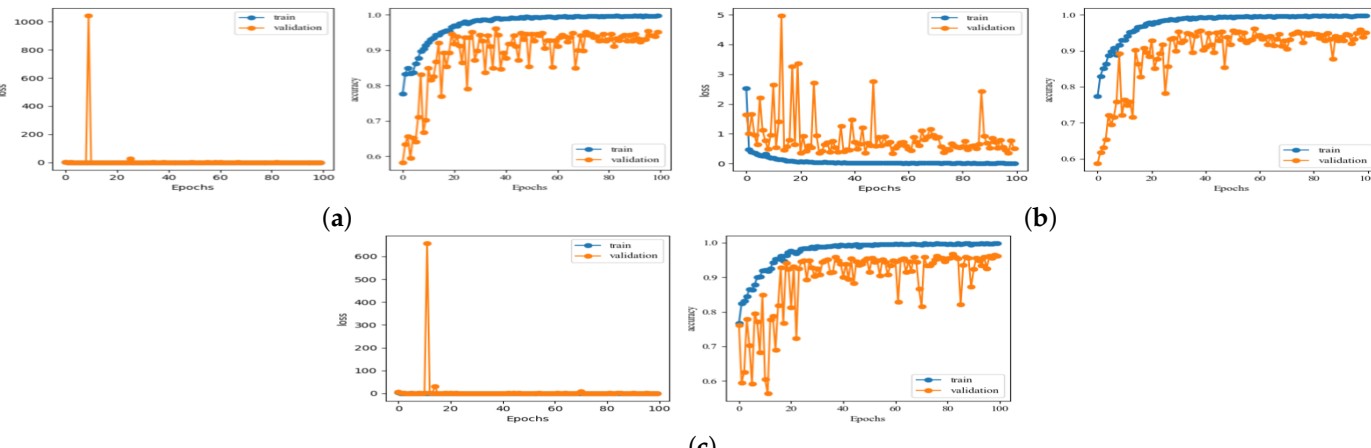

**Figure 9.** SSS target recognition results with DenseNet series networks: (**a**) DenseNet201 loss and accuracy curves; (**b**) DenseNet121 loss and accuracy curves; (**c**) DenseNet169 loss and accuracy curves.

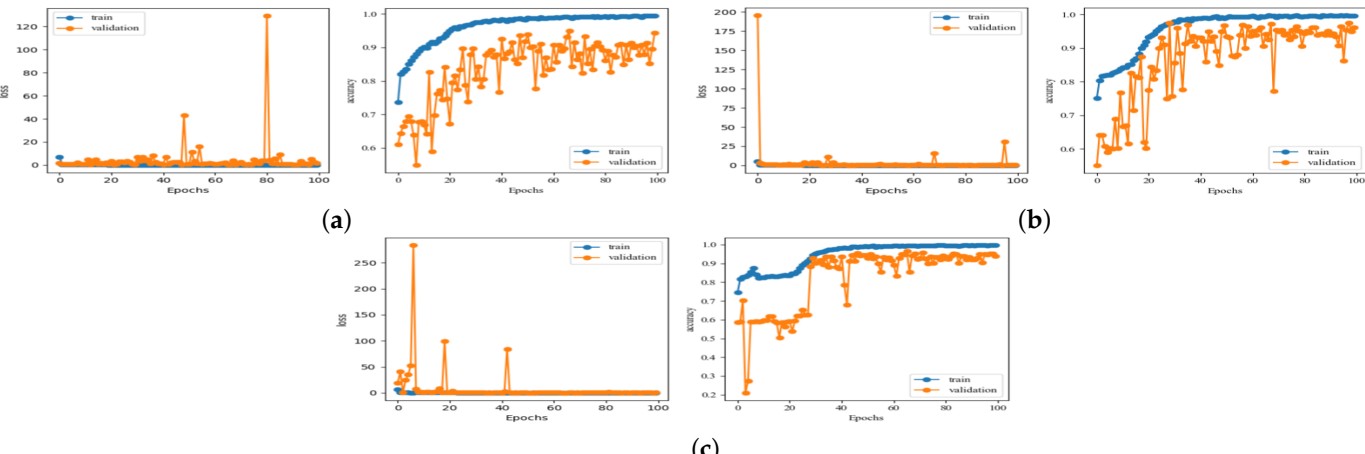

**Figure 10.** SSS target recognition results of ResNet series networks: (**a**) ResNet50 loss and accuracy curves; (**b**) ResNet101 loss and accuracy curves; (**c**) ResNet152 loss and accuracy curves.

As Table 3 shows, the model structures of VGGNet16 and VGGNet19 are simple and easy to train; however, their actual recognition performance is not ideal, and their recognition accuracy is only 92.56%. As shown in Figure 11, during the training process of VGGNet16 and VGGNet19 models, the recognition accuracy curves exhibit instability and jitter, as shown in Figure 11a,b. In particular, the VGGNet19 network shows a downward trend with an increasing number of epochs, and the final recognition rate is only 60%, as shown in Figure 11b. DenseNet (201, 121, 169) and ResNet (152, 101, 50) network models have similar recognition accuracies, and the ResNet101 has the optimal recognition accuracy, which is equal to 96.41%. At the same time, the recognition accuracy of the ResNet network layer decreases slightly from 96.41% for 101 layers to 95.13% for 152 layers, and the recognition accuracy of the DenseNet network layer decreases from 96.15% for 169 layers to 95.02% for 201 layers. This is because as the number of network layers increases, the features extracted by the network tend to be more abstract and detailed. As the SSS image has a large noise interference, the extracted detailed features include noise features, which directly affect the recognition performance of the network.

**Table 3.** Comparison of different DL models for SSS image object recognition.

| Methods | Optimal OA (%) |
|---|---|
| DenseNet201 | 95.02 |
| DenseNet121 | 95.13 |
| DenseNet169 | 96.15 |
| ResNet50 | 91.28 |
| ResNet101 | 96.41 |
| ResNet152 | 95.13 |
| VGGNet16 | 92.56 |
| VGGNet19 | 92.56 |

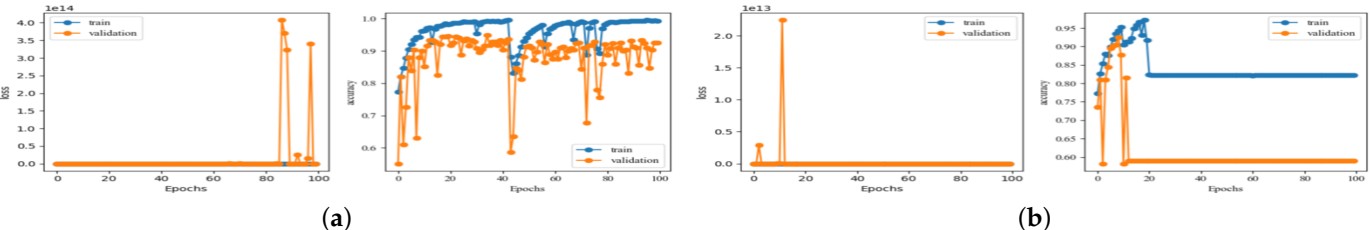

(**a**)                              (**b**)

**Figure 11.** SSS target recognition results of VGGNet series networks: (**a**) VGGNet16 loss and accuracy curves; (**b**) VGGNet19 loss and accuracy curves.

(2) Comparison of methods for the classification of SSS images

A few SOTA methods for SSS target recognition have been proposed in [34–36]. These methods use different data enhancement methods to increase the number of training data samples through semi-synthetic data [36] and improve the learning and recognition capabilities of the network, which increases the recognition accuracy significantly. Table 4 shows that using synthetic data and VGG19Net fine-tuning network method, the highest target recognition accuracy of 97.76% can be achieved. The recognition accuracy of other methods is reduced to varying degrees due to the insufficient number of network layers.

**Table 4.** Comparison of different methods for SSS image object recognition.

| Methods | Optimal OA (%) |
|---|---|
| Shallow CNN [34] | 83.19 |
| GoogleNet [35] | 91.86 |
| VGG11 fine-tuning + semisynthetic data [24] | 92.51 |
| VGG19 fine-tuning [25] | 94.67 |
| VGG19 fine-tuning + semisynthetic data [25] | 97.76 |
| SPDRDL [36] | 97.38 |
| Our Methods | 98.72 |

As the time consumed by the synthetic data increases according to the complexity of the synthetic model, the time consumed for processing the synthetic data is relatively large. It takes a considerable amount of time to prepare the data before training the recognition network. In underwater recognition tasks that do not require a long training time, it is not appropriate to use synthetic data to train a network for tasks with low time complexity, because the time consumed by the synthetic data will be calculated as a part of the recognition time. The staged attention transfer recognition method proposed in this paper can significantly improve the target recognition accuracy rate to 98.72% through the target domain data attention enhancement method without any time loss caused by additional sample generation. This recognition performance is the best compared to other methods, showing an increase of 1%.



(3) Comparison of different backbones for the classification of SSS image.

The ablation experiment method is used to pre-train a variety of backbone networks using ImageNet data, and the transfer recognition method is used to fine-tune the parameters of the head network using SSS data to analyze the recognition performance. In order to fully analyze the positive effect of transfer recognition on SSS target recognition, we supplemented AlexNet, GoogleNet and other shallow network transfer recognition methods based on the above traditional network model (ResNet, DenseNet, VGGNet) for comparative analysis. Figures 12–14 show the recognition rate and loss curves of the partial recognition network, and Table 5 shows the optimal recognition rate results with different backbone network transfer recognition methods.

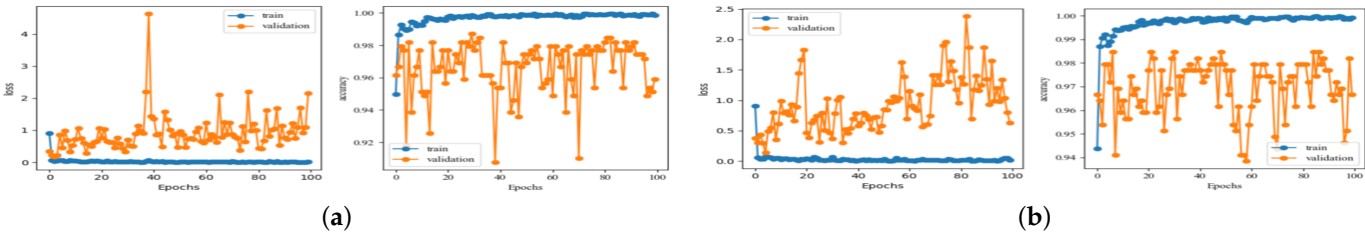

**Figure 12.** SSS target transfer recognition results with ResNet series networks: (**a**) ResNet152 loss and accuracy curves; (**b**) ResNet101 loss and accuracy curves.

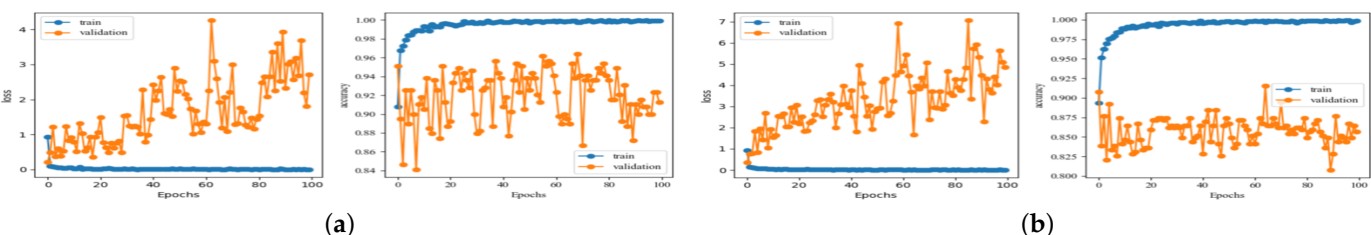

**Figure 13.** SSS target transfer recognition results with DenseNet series networks: (**a**) DenseNet121 loss and accuracy curves; (**b**) DenseNet169 loss and accuracy curves.

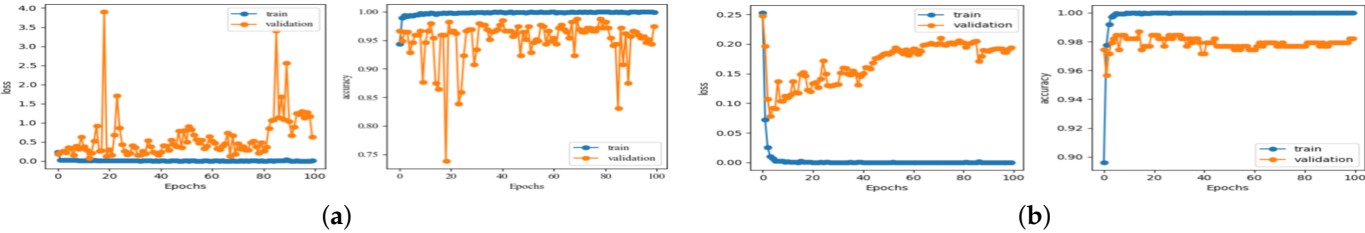

**Figure 14.** SSS target migration recognition results with VGGNet16 network and the proposed method. (**a**) VGGNet16 loss and accuracy curves; (**b**) proposed method loss and accuracy curves.

Table 5 shows that different backbone networks used to identify SSS targets produce different results: VGGNet16, ResNet152, and Resnet101 have a recognition accuracy rate higher than 98%, while ResNet18 and DenseNet169 have relatively low recognition accurate rates of only 91.86% and 91.54%, respectively. This is because the number of network layers is small for ResNet18: there are only 18 layers that are not sufficient for feature extraction. For DenseNet121, DenseNet169 and DenseNet201, the values of OA are 96.41%, 91.54% and 89.23%, respectively. It can be concluded that as the number of network layers increases, the recognition accuracy rate shows a decreasing trend. This phenomenon is caused by a difference between the training data of the backbone network and the SSS data. As the number of network layers increases, the network extracts more detailed and abstract features. The SSS data have a significant amount of noise interference. This noise exists in the details of the image. The deeper network extracts the noisy features during the

extraction of the detailed features. These noisy features have a negative impact on the target recognition performance. Therefore, as the network structure becomes more complex and the number of network layers increases, the recognition rate tends to decline.

**Table 5.** Comparison of different backbones for SSS image object recognition.

| Backbone Networks | Optimal OA (%) |
|---|---|
| ResNet18 | 91.86 |
| ResNet50 | 93.5 |
| ResNet152 | 98.21 |
| ResNet101 | 98.46 |
| DenseNet121 | 96.41 |
| DenseNet169 | 91.54 |
| DenseNet201 | 89.23 |
| VGGNet16 | 98.46 |
| VGGNet19 | 94.67 |
| AlexNet | 94.14 |
| GoogleNet | 94.46 |
| Proposed | 98.72 |

The method proposed in this paper comprehensively considers the influence of the number of network layers and network structure on the recognition performance and uses ResNet152 as the backbone network to learn the network parameters and identify the SSS targets. It obtains the best recognition accuracy rate of 98.72%. At the same time, Figures 12–14 show that the recognition rate curves of the existing methods have a large variance, whereas the variance of the recognition rate curve of the proposed method is relatively small. This behavior indicates that the recognition performance of the method proposed in this paper is more stable.

(4) Ablation experiments for backbone network trained on different data sets.

In order to verify the DL strategy problem, this paper uses SAR, ImageNet, and SSS data to conduct ablation experiments on the sub-network structure of the backbone network. Table 6 shows the experimental results.

**Table 6.** Ablation experiments for backbone network trained on different data sets.

| Backbone / Datasets | Conv1 Conv2_x | Conv3_x Conv4_x | Conv5_x CAMCL LAMCC | Optimal OA (%) |
|---|---|---|---|---|
| | ImageNet | ImageNet | SSS | 58.97 |
| ImageNet/SAR/SSS | SAR | SAR | SSS | 77.95 |
| Datasets | ImageNet | SAR | SSS | 96.92 |
| | SAR | ImageNet | SSS | 98.72 |

Table 6 shows that the recognition performance of the backbone networks trained only with ImageNet and SAR data set is poor, achieving OA values of only 58.97% and 77.95%. The recognition performance of the backbone network trained with two data sets is better, and the OA of the training strategy (ImageNet+SAR) is 1.8% lower than that proposed in this paper (SAR+ImageNet).

(5) CAMCL and LAMCC ablation experiments.

In order to verify the performance of the CAMCL and LAMCC attention fusion methods, this paper conducts attention mechanism ablation experiments. Experiments were carried out on three methods: only CAMCL, only LAMCC, and CAMCL+ LAMCC with ResNet152 as the backbone network.

Table 7 shows that the optimal OA values for only CAMCL and only LAMCC are 97.69% and 97.18%, respectively. The optimal OA value for CAMCL+ LAMCC is 98.72%, signifying an increase in the recognition rates by 1.03% and 1.54% respectively. Therefore, the use of CAMCL and LAMCC attention fusion mechanisms is helpful to improve recognition performance.

**Table 7.** CAMCL and LAMCC ablation experiments results.

| Backbone | Attention Module | CAMCL | LAMCC | Optimal OA(%) |
|---|---|---|---|---|
| ResNet152 | | ✓ | | 97.69 |
| | | | ✓ | 97.18 |
| | | ✓ | ✓ | 98.72 |

(6) Application of FLS Target Recognition.

The method proposed in this paper can not only recognize SSS targets but can also be applied to FLS target recognition. In order to verify the effectiveness of the proposed method with respect to a higher number of target recognition categories, this paper uses Forward Looking Sonar (FLS) data for experiments. Among them, the total number of samples is 3192, the number of target categories is 10, the number of training sets is 2231, and the number of test sets is 961. Figure 15 shows the data sample diagram and Table 8 shows the experimental results.

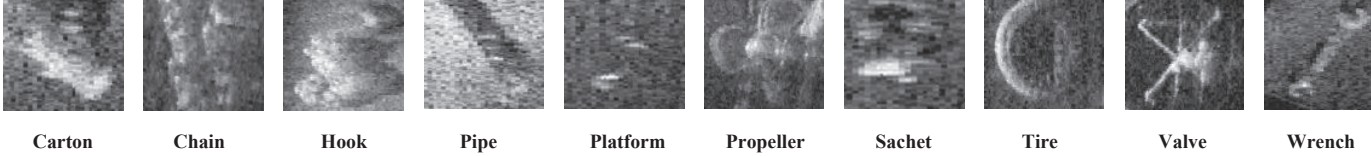

| **Carton** | **Chain** | **Hook** | **Pipe** | **Platform** | **Propeller** | **Sachet** | **Tire** | **Valve** | **Wrench** |

**Figure 15.** Forward-looking sonar dataset samples diagram.

**Table 8.** Comparison of different DL models for FLS image object recognition.

| Methods | Optimal OA (%) |
|---|---|
| DenseNet201 | 89.07 |
| DenseNet121 | 88.87 |
| DenseNet169 | 89.91 |
| ResNet50 | 89.49 |
| ResNet101 | 88.14 |
| ResNet152 | 88.03 |
| VGGNet16 | 90.63 |
| VGGNet19 | 85.22 |
| Proposed | 93.96 |

Table 8 shows that the recognition rates of the DenseNet201, DenseNet121 and DenseNet169 are equal to 89.07%, 88.87% and 89.91%, respectively, which are slightly higher than those obtained using the ResNet series algorithms. The recognition rates of DenseNet and ResNet algorithms are above 88%. VGGNet16 achieves the best OA = 90.63% among the reference algorithms, and VGGNet19 performs the worst with OA = 85.22%. The method proposed in this paper is the best among all the comparison algorithms, which shows its good FLS target recognition performance.

## 4. Conclusions and Future Work

An important part of the sonar system that provides accurate target perception information is the SSS underwater target recognition. In this study, a new 3MATR model was provided for SSS target recognition, using the ResNet152 network as the backbone network of the model. The strategy of phased transfer, ImageNet and SAR data were collectively utilized to learn the parameters of the backbone network. In addition, the CAMCL and LAMCC attention mechanisms were used to perform the final and accurate recognition of SSS data in the head network. The proposed model was analyzed from different aspects and its performance was compared with many existing identification algorithms. The following conclusions were obtained:

1. The proposed multi-modal multi-channel multi-position attention mechanism transfer recognition method effectively improved the SSS target recognition accuracy. Its basic principle was to train different stages of the network through ImageNet, SAR, and SSS data sets to enhance the anti-noise performance of the network against noisy backgrounds, and simultaneously add an attention mechanism to extract key features in the SSS training stage.

2. The 3MATR model provided advantages in SSS target recognition, which resulted in significantly better recognition rates than other backbone networks. Its transfer recognition performance was higher than other transfer learning target recognition methods.

The limitation of this method was the difference in data distributions for different modalities in the transfer model. In future work, we envisage overcoming the differences in data distribution by improving the model and further improving the recognition performance.

**Author Contributions:** Conceptualization, J.W. and H.L.; Methodology, J.W. and G.H.; Software, J.W.; Validation, J.W. and H.L.; Formal analysis, J.W. and C.L.; Investigation, J.W. and Y.W.; Resources, H.L.; Data curation, H.L.; Writing—original draft preparation, J.W.; Writing—review and editing, J.W. and H.L.; Visualization, J.W.; Supervision, G.H. and H.L.; Project administration, G.H. and H.L.; Funding acquisition, H.L. All authors have read and agreed to the published version of the manuscript.

**Funding:** This research was funded by National Natural Science Foundation of China under grant nos. U1906218 and 2021YFC3101803; in part by Natural Science Foundation of Heilongjiang Province, under grant no. ZD2020D001; and in part by key areas of research and development plan key projects of Guangdong Province under grant no. 2020B1111010002.

**Institutional Review Board Statement:** Not applicable.

**Informed Consent Statement:** Not applicable.

**Data Availability Statement:** Access to the data will be available at https://toscode.gitee.com/wangjian19870118/ssd-dataset.git (accessed on 16 November 2021).

**Acknowledgments:** We would like to thank Guanying Huo, L-3 Klein Associates, EdgeTech, Lcocean, Hydro-tech Marine, and Tritech for their great support for providing the valuable real side-scan sonar images.

**Conflicts of Interest:** The authors declare no conflict of interest.

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
