# Peer review of "Multi-Mode Channel Position Attention Fusion Side-Scan Sonar Transfer Recognition"

_electronics, doi:10.3390/electronics12040791_

Round 1

Reviewer 1 Report

The authors have proposed a multi-channel and multi-position attention mechanism for SSS target recognition. Precisely, it consists of CAMCL and LAMCC, and they utilize SAR, ImageNet, and SSS datasets to train the front, middle, and head of the network, respectively. In addition, the authors conducted multiple experiments to support their claims. However, some concerns remain regarding the proposed method.

1. Why did you use SAR data to train the front of the backbone and ImageNet data to train the middle part of the backbone? Ablation studies in various settings should be needed. For instance, i) (SAR+ImageNet) all layers of the backbone, ii) (SAR only) all layers of the backbone, iii) (ImageNet only) all layers of the backbone, iv) (SAR) middle of the backbone+ (ImageNet) front of the backbone

2. Did you freeze all the other layers of the backbone when training the front of the model with SAR data? How about ImageNet and SSS datasets to learn the middle part of the backbone and the head and Conv5, respectively?

3. Ablation studies for each CAMCL and LAMCC should also be performed. 

4. Did you directly acquire the SSS dataset? If not, you need to add a reference.

5. On page 8, line 179: "Table 1" is a typo, and the appropriate table w.r.t the specific data allocation for each category is missing.

6. Figures 7, 9, 12, and 14 and each corresponding caption should be on the same page.

7. The current experiment was performed on only one SSS dataset. Aren't there more diverse side-scan sonar datasets? It is necessary to perform additional experiments on other datasets.

8. On page 15, lines 264 to 267: Is the performance drop in DenseNet169 caused by an increase in the number of layers? In Table 2, DenseNet201 does not have a significant performance drop compared to DenseNet169 and is similar to DenseNet121. It seems to be needed experimental results from DenseNet201.

9. On page 16, line 312: Access to the data is not available at the link. 

Reviewer 2 Report

In this manuscript, the authors propose a side-scan sonar target recognition model, which is interesting to read. However, I would like to suggest a minor revision before acceptance. The title of the manuscript contains the repeated word "multi," and it could be rewritten. The CAMCL and LAMCC parts contain many mathematical descriptions without proof of correctness or proper citations.

Round 2

Reviewer 1 Report

Additional experiments have been added to verify the ablation study and hyperparameters of the proposed methods.

The detailed explanation of the proposed methods has been explained clearly.

I am satisfied with the revised manuscript.